# A Real-Life Action toward the End of HIV Pandemic: Surveillance of Mother-to-Child HIV Transmission in a Center from Southeast Romania

**DOI:** 10.3390/jcm11175020

**Published:** 2022-08-26

**Authors:** Manuela Arbune, Alina Mihaela Calin, Alina Viorica Iancu, Caterina Nela Dumitru, Anca Adriana Arbune

**Affiliations:** 1Clinical Medical Department, Medicine and Pharmacy Faculty, Dunarea de Jos University of Galati, 800008 Galati, Romania; 2Infectious Diseases Clinic Department I, Clinical Infectious Diseases Hospital Galati, 800179 Galati, Romania; 3Clinic Surgical Department, Medicine and Pharmacy Faculty, Dunarea de Jos University of Galati, 800008 Galati, Romania; 4Obstetrics and Gynecology Clinic, Emergency County Hospital Sf. Apostol Andrei, 800578 Galati, Romania; 5Morphological and Functional Sciences Department, Faculty of Medicine and Pharmacy, Dunarea de Jos University of Galați, 800008 Galati, Romania; 6Clinic Laboratory Department, Clinical Infectious Diseases Hospital Galati, 800179 Galati, Romania; 7Pharmaceutical Sciences Department, Medicine and Pharmacy Faculty, Dunarea de Jos University of Galati, 800008 Galati, Romania; 8Neurology Clinic, Fundeni Clinical Institute, 022328 Bucharest, Romania

**Keywords:** HIV, antiretroviral, mother-to-child transmission, perinatal infection, perinatal deaths, birth defects, seroreverter

## Abstract

Preventing mother-to-child HIV transmission is a strategy to eliminate new infections to move toward a world free of HIV/AIDS. The aim of this study is to assess the effectiveness of the perinatal infection prevention program in a single center from southeast Romania. Newborns of HIV-positive mothers from 2005 to 2020 were followed-up until the age of two in a retrospective study. The transmission rate from HIV-positive mothers to living children was zero, but neonatal mortality, preterm birth and birth defects were still high. The peculiarity of our study is the high proportion of mothers with a nosocomial pattern of HIV transmission. Intensifying the efforts for accurate implementing the interventions for the prevention of mother-to-child HIV transmission, a long time follow-up for HIV-exposed uninfected children and new research on related HIV pregnancies are necessary to reach the objective of a new generation free of HIV.

## 1. Introduction

After four decades of evolution of the HIV/AIDS epidemic globally, medical and technological advances have led to an understanding of the specific pathogeny, clinic and therapeutic aspects of HIV infection. Although most HIV infections are common in adults, almost 2 million children under the age of 15 continue to be diagnosed with HIV each year, 90% of them with perinatal transmission [1].

Perinatal HIV infection could be transmitted from a positive woman to her baby, during pregnancy, childbirth or breastfeeding. Most infections, in the absence of antiviral treatment, occur during birth, with a risk of transmission between 15–25% and an additional risk of 15–20% if the baby is breastfed. In the case of mothers receiving antiretroviral treatment and other preventive interventions, the risk of transmission could decrease below 5% [2]. These interventions include measures for safe births, infant feeding formula and preventive antiviral medication for perinatal exposed infants [2,3]. In the UK and Ireland, the perinatal transmission rate is even lower, at 0.3% [4].

The World Health Organization has set the goal to eliminate mother-to-child transmission (PMTCT) by 2030 [1]. Providing antiretroviral medication to all pregnant women, regardless of the CD4 value or stage of the disease, decreased to 54% of new pediatric infections from 2010 to 2020, but HIV diagnostic failure in fertile women (15 to 49 years) is still an obstacle to the elimination of pediatric HIV cases [1,2].

Antiretroviral medications have unquestionable benefits in preventing perinatal HIV transmission [5,6,7]. Although the effective treatment constitutes the most successful strategy for preventing onward transmission of infections and the expansion of the HIV epidemic, including mother-to-child transmission, the risk of genotoxicity is incompletely known. Zidovudine was, for a long time, the only antiretroviral approved for use during pregnancy, although it is less prescribed nowadays due to adverse events [8]. Afterwards, other drugs, such as Lopinavir and Raltegravir, were registered for use during pregnancy.

In Romania, PMTCT is supported within a national prevention program, including the recommendation of HIV screening during pregnancy, assessment, monitoring, and antiretroviral treatment of pregnant women, planning caesarean section delivery, infant feeding formula, and antiretroviral prophylaxis of the newborn up to 6 weeks after birth. Safe birth means the minimum exposure of the fetus to HIV during birth, indicating elective caesarean section when there is no certainty about the undetectable HIV viral level of the mother at birth [9]. A peculiarity of the HIV epidemic in Romania is the large number of nosocomially infected infants from 1988–1990, mainly involving the institutionalized children and hospitalized children who received blood micro transfusions or parenteral treatments. Some of these children, who constituted the “pediatric cohort”, survived, and reached the fertile age, having children in their turn. Previous exposure of pregnant women to multiple antiretroviral treatment regimens, during childhood and puberty, could accumulate metabolic, endocrine and vascular side effects that may influence pregnancy. Therapeutic fatigue with a risk of poor adherence are additional difficulties for the PMTCT [10].

The aim of this study is to evaluate the effectiveness and limits of PMTCT in a center from southeastern Romania. The main objective of the study was to find out the prevalence of HIV perinatal infections. We investigated the neonatal mortality and the frequency of birth defects in children with HIV-positive mothers as secondary objectives.

## 2. Materials and Methods

We conducted a retrospective study from 2005–2019 on children monitored for at least the first 2 years of life, born to HIV-positive mothers recorded in the HIV/AIDS Department of the Clinical Hospital for Infectious Diseases in Galati, Romania. The interventions for PMTCT were classified into 3 stages (Figure A1):Prenatal interventions for mother HIV diagnosis, evaluation and treatment.Birth decision for procedures of vaginal or caesarean section.After birth intervention for the clinical evaluation of newborns, feed them with formula, antiretroviral medication for six weeks, subject them to virologic and serologic monitoring for at least 18 months.

We evaluated the relationship between pregnancy and the HIV diagnosis of positive pregnant women: before pregnancy, during pregnancy or at birth. HIV diagnosis in pregnant women was the result of HIV screening during pregnancy or at birth. Pregnancy identification in fertile women already known to have HIV is part of the routine monitoring. The assessment of maternal risk factors for perinatal HIV transmission considered the duration of HIV diagnosis, the stage of HIV infection (AIDS/non-AIDS), antiretroviral treatment during pregnancy, the level of LCD4 immunity and the level of viremia after 28 weeks of pregnancy. The PCR-HIV evaluation at birth is not available in our center.

The type of delivery, vaginally or by caesarean section, was notified in each case, depending on obstetrical decisions or the nature of the mother’s HIV diagnosis and pregnancy. The preferred delivery intervention is the elective caesarean section at 38 weeks of pregnancy, while vaginal delivery is mostly related to late-presenting cases.

We evaluated specific interventions for newborns of mothers with an HIV infection: fed exclusively with milk formula and given pediatric antiretroviral drugs in the first hours of life, according to current protocols [3,11]. Along the course of the study, the timeline of prophylactic antiretroviral guidelines and recommendations were changed, and a particular criterion for choosing the treatment was not available. As a rule, the associations of newborn antiretrovirals depended on the risk of transmission, which was considered as related to the level of maternal viremia, type of delivery, gestational age and birth weight. Zidovudine (AZT), zidovudine and lamivudine (AZT + 3TC), or zidovudine, lamivudine and nevirapine were used (AZT + 3TC + NVP) [11]. HIV viremia was assessed at 14–21 days, 4–6 months and 18 months, by Roche COBAS TaqMan assays or Abbott LCx quantitative assays for HIV-1 RNA HIV-1 techniques, considering the detection limit of 40 copies/mL. The serological evaluation was performed at the age of 18–24 months, with Greenscreen HIV Ag/Ab and Murex Ag/Ab tests. According to the case definition, children exposed to perinatal HIV were seroreverter if the viral load was undetectable, and the serological tests were negative at the end of the follow-up, during the first 18 months of life [3].

Demographics were assessed for both the mother and child.

The data collected from the newborns were the Apgar score, anthropometric data (weight, length, cranial circumference) and gestational age. Preterm birth considered a gestational age of less than 37 weeks [12]. The general screening protocol of the newborn supported by the Romanian Ministry of Health included tests for phenylketonuria and congenital hypothyroidism, congenital malformations by pulse oximetry, and hearing loss by otoacoustic emission test. We notified the type of defects clinically highlighted at birth, supplemented by other investigations, depending on the individual situation [13,14]. In the first 6 weeks, additional rounds of transfontanellar, cardiac and hip echography were performed.

The statistical analysis was based on data in MS Office Excel 2021 (Microsoft, Redmond, WA, USA) on Windows11. Mean, median, minimum and maximum values were used for continuous variables. Frequencies and percentages were used to summarize categorial data. Differences between the proportions of categorial variables were evaluated by the Chi-squared test, and *p*-values under 0.05 were considered statistically significant.

The study was conducted according to the guidelines of the Declaration of Helsinki and approved by the Ethics Committee of Infectious Diseases Clinical Hospital, Galati, Romania.

The informed consent for processing the medical data for research purposes was signed as part of a process of accepting the medical procedures, according to the hospital’s regulation.

## 3. Results

A group of 114 children, have been born from 2005 to 2019, by 87 HIV-positive Caucasian women, 65% of them experiencing their first pregnancy. The age of HIV-positive mothers at birth ranged from 16 to 38 years, with a mean of 24.57 ± 4.59 years. Formal education of most pregnant women was limited to 8 years (54.4%), but around a third of them have a formal education of less than 4 years. By marital status, 46% of mothers were married, 40% were unmarried couples and 14% were single mothers. The HIV status of the partners was 54% negative, 34% positive and 12% unknown (Table 1).

Multiple experiences with antiretrovirals characterized 75% (65/87) of the pregnant women, 63 of them belonging to the Romanian HIV pediatric cohort. The diagnosis of HIV was known before pregnancy in 76% (87/114) of births. The median duration of HIV diagnosis was 11 years, with variations ranging from 1 to 28 years. The HIV screening program in pregnant women allowed for diagnosis during pregnancy in 10% of mothers, but 9% of pregnancies were not monitored, and HIV diagnosis was late to delivery. The history of AIDS stage, according to CDC classification, was reported in 57% of pregnant women, but no opportunistic disease was developed during the pregnancy.

Coinfection with HBV, defined as HBs-Ag and/or HBc-Ab, was found in 28%, stating that HBV-DNA was not routinely assessed. Antiretroviral treatment was provided to 82.5% (94/114) of pregnant women, of whom 21% (24/114) began the treatment after the first trimester. Prenatal antiretroviral intervention was used in 20 pregnancies, due to the delayed HIV diagnostic until birth or non-adherence, and was provided after the first trimester in 24 cases, consisting of zidovudine/lamivudine and lopinavir/ritonavir. The current regimens of antiretrovirals in treatment-experienced patients were continued along the course of the pregnancy. The assessment in the last trimester of pregnancy has evidenced LCD4 ranging between 8 and 1543/mm^3^, with a median value of 548/mm^3^ and undetectable HIV viral loads in 59% of cases (under 400 copies/mL). Fetal antiretroviral exposure during pregnancy involved combinations of protease inhibitors (71%), non-nucleosides reverse-transcriptase inhibitors (7.9%) and integrase inhibitors (8.8%) (Table A1).

Two pairs of twins were identified between the perinatal HIV-exposed newborns. The M/F ratio was 1.03 (58/56). The mean gestational age was 37.21 ± 2.20 weeks, mean Apgar score 8 ± 1.02, mean birth weight 2739 ± 512.03 g, mean length 48.26 ± 3.32 cm (Table 2).

A proportion of 28.07% of premature infants and 7.89% of newborns were diagnosed with low birth weight for their gestational age.

Prophylaxis measures included in the national program were performed as follows: antiretroviral treatment (ARVT) of the mother, at least in the last trimester of pregnancy (82.45%), caesarean section delivery (89.47%), newborn formula feeding (98.25%) and antiretrovirals prophylaxis to the newborn, from the first 6–12 h of life (94.74%) (Figure 1).

No procedures of forceps extraction were used.

Neonatal respiratory distress was reported in 23.68% (27/114) of perinatal HIV-exposed infants. Antiretroviral pediatric treatment included zidovudine and was considered according to the protocols, the gestational age, the birth weight and maternal virologic state (Table A2). Clinical and ultrasound postnatal evaluation revealed multiple musculoskeletal, neurological, cardiovascular, urogenital or abdominal wall defects (Table A3).

At least one birth defect was found in 47.36% (54/114) of newborns, 33 of whom had multiple defects. However, most cases (38.58%) were minor anomalies, according to the current European Surveillance of Congenital Anomalies (EUROCAT) classification [10]. Angiomas were the most common birthmark in 27.19% (31/114) of newborns, but no case had any indications for surgery. Musculoskeletal defects were found in 24% of newborns, represented by dislocation/dysplasia of the hip (6/114), equine leg (14/114), valgus metatarsus (3/114) and congenital torticollis (1/114). Other minor defects had neurological localization (subarachnoid/arachnoid/choroid plexus cysts, microcephaly, lenticulostriate vasculopathy, occult *spina bifida*, *corpus callosus* dysgenesis), cardiology (persistence of the duct of the artery, interatrial septal aneurysm, *situs solitus*, cardiomyopathy), cryptorchidism and hydrocele (Table A3). Phenotypic features, complemented by imaging results, identified two complex syndromes in children living with oculo-atrial-vertebral syndrome and Dandy–Walker syndrome [15].

After excluding minor anomalies and unspecified anomalies, major congenital anomalies were notified in 8.7% (10/114) of newborns: 6.13% cardiovascular, 1.75% urogenital and 0.87% abdominal wall defects (Table 3).

Analysis of birth defects and exposure to each antiretroviral drug did not find significant correlations (Table A4). Furthermore, we did not find correlations of musculo-skeletal defects and vaginal birth (*p* = 0.0186), between birth defects and the mother belonging to the HIV Romanian cohort (*p* = 0.348) or the use of antiretrovirals during the first trimester of pregnancy (0.707).

The rate of neonate deaths among HIV-exposed newborns was 2.63% (3/114). None of these cases had HIV-RNA or proviral HIV-DNA testing. A newborn with gastroschisis died in the third day after surgery; the HIV diagnosis of his mother was at delivery, and ge had no intrauterine exposure to antiretrovirals. The other two deaths were caused by cerebral haemorrhages, in the context of prematurity; the mothers of both newborns were twin sisters.

Seroreverter status was found after serologic and virologic evaluation at the age of 18–24 months in all 111 surviving neonates with HIV-positive mothers.

The rate of perinatal HIV transmission after fifteen years (2005–2019) was zero, after early neonatal deaths are excluded, with unavailable HIV virologic tests.

## 4. Discussion

### 4.1. Local Condition of Perinatal Transmission

The elimination of perinatal transmission secures a new AIDS-free generation of children. The yearly number of Romanian newborns in the whole country with perinatal HIV infections has varied in the last 10 years from 6 to 28 cases, meaning between 0.7% and 4.6% of the new cases yearly, distributed by the means of transmission [10]. The perinatal transmission rate reported in other Romanian centers varied between 3% and 7% [16,17,18]. The result of our study, was “zero maternal-fetal infections”, strengthening the regional hope of eradicating HIV infections.

### 4.2. Peculiarities of HIV-Positive Mother Survivors from the Pediatric Cohort

A peculiarity of our study is the preponderance of mothers with pediatric HIV infections associated with the medical care acquired in the first years of life, who survived and gave birth to a new generation of children. Along with the positive emotional aspects, the management of these cases brings to attention two major challenges. Low education, social discrimination, psychological frustrations or poverty, as well as AIDS- and non-AIDS-associated diseases, are more frequent burdens for the young females with HIV from the pediatric cohorts. Therapeutical fatigue, poor adherence and virologic failure are often difficult to manage. Other queries with incomplete answers. What is the influence of HIV infections, maternal antiretroviral medication and caesarean sections on the preterm-birth and birth mortality of the mother and child? How efficient is the antiretroviral prophylaxis in newborns of mothers with long-term therapeutic experiences and possible HIV resistance mutations? What are the implications of antiretroviral therapy on embryo–fetal development and the presence of birth defects?

### 4.3. The Impact of HIV on Maternal and Neonatal Mortality

HIV-associated global maternal mortality has declined over the past two decades, although in some disadvantaged regions, the risk of maternal deaths in HIV-positive women remains more than five times higher than in uninfected women [19,20,21]. No maternal deaths were recorded in our center. At the same time, WHO statistics for Romania indicate zero maternal deaths associated with HIV and an overall maternal mortality rate with a declining trend, up to 5.2/100,000 births [16]. The perinatal mortality rate declined worldwide, and the discrepancy between children born to HIV-positive and HIV-negative mothers has been narrowed [22]. During the study period, the general perinatal mortality rate decreased in Romania from 1.06 to 0.57% [19]. In our study, 3 newborns died in the first 6 days of life, meaning 2.6% perinatal mortality associated with HIV exposure. This is higher than other recent reports, as the cohort study from the UK and Ireland showed, with 0.41% mortality rate [4].

### 4.4. The Impact of Preventive Interventions on HIV-Exposed Children

Pediatric antiretrovirals were provided in 108 of 111 newborns, with Zidovudine alone or in combination with Lamivudine ± Nevirapine. The effectiveness of these medications for the children of mothers from the pediatric cohort may be limited due to viral resistance mutations acquired by the mothers’ therapeutic experiences. Little is known on antiviral administration in children of mothers infected perinatally or during early childhood, with a huge therapeutic experience. A study of 11 pregnant women who had been infected perinatally with HIV found mutations in resistance to zidovudine and lamivudine. However, all their 21 babies were given prophylaxis with these antivirals. Although maternal viremia at birth was detectable in half of the cases, all children were uninfected [23].

Preterm delivery (<37 weeks) could be an additional factor for HIV transmission, although the risk is minimal for women with predelivery viral loads less than 1000 copies/mL [8,11,24]. The rate of preterm births for HIV-exposed babies in our study was 28.07%, which is 2.7 times higher than the average of 10.6% from global European data [25]. Increased rates of preterm births related to mothers’ HIV infection have been also reported by other studies: 22.4% in another Romanian center, 22.6% in Italy, 20% in the USA and 21.7% in Brazil [26,27,28,29].

Although an elective caesarean section is recommended at 38 weeks of pregnancy, the decisions to achieve it earlier was related to the premature beginning of labor, premature rupture of the membranes or evidence of fetal distress. Considering the high rate of detectable or unavailable viral loads for mothers at birth, a caesarean section is chosen in Romania as a safer procedure than vaginal delivery due to the risk of mother-to-child HIV transmission, [17].

Zero maternofoetal infections in living babies were achieved in our study, even if the cascade of care was not fully accomplished. Systematically feeding babies with baby formula had a major contribution, considering that 5–20% of mother-to-child transmitted infections are through breastfeeding [3]. The weakest indicator is the detectable HIV viral load in 41% of pregnant women, due to either late HIV diagnosis during pregnancy, or non-adherence when the HIV diagnosis was known. The reasons for non-adherence could be the fear of disclosing their HIV diagnosis, the fear that antiretrovirals harm the baby, or therapeutical fatigue. The resistance tests are not usually available in our site, but the acquired resistance to antiretroviral drugs is considered in cases of virologic failure. Caesarean section, as a usual indication in 92% of births, could have decreased the risk of transmission related to the detectable viral load. Antiretrovirals were provided to all the alive babies in the first hours of life, but were discontinued in three cases due to toxicity.

Congenital anomalies can be caused by genetic, environmental or other risk factors, but almost half of the cases cannot be related to a specific cause. Most trials of in utero exposure to antiretrovirals have not shown significant risks for birth defects, but there are studies that show higher risks for some specific drugs. The Antiretroviral Pregnancy Registry (APR) reported 2.87% birth defects, compared to the metropolitan Atlanta congenital program’s 2.72% and 3.56% of live births, observed in a cohort of 2527 newborns of HIV-infected mothers [30,31,32].

Pediatric AIDS Clinical Trials Group 219 and 219 C protocols enrolled 2202 HIV-exposed children before one year of age, reporting a higher prevalence of birth defects of 5.3%, with a higher rate of heart defects when zidovudine was administered in the first trimester of pregnancy [33]. The French Perinatal Cohort included 13,124 live births from 1994 to 2010, of which 42% were exposed to ART in the first trimester of pregnancy. A prevalence of 4.4% of birth defects was reported for children prospectively followed up until the age of two, according to the EUROCAT classification, but with no evidence of associations with antiviral exposure to lopinavir, ritonavir, nevirapine, tenofovir, stavudine, or abacavir. Associations between zidovudine exposure in the first trimester with congenital heart defects, or to efavirenz with neurological defects, were not significant [34].

The birth defect rate in our study, according to the EUROCAT classification, was 8.7%, 1.52 times higher in pregnant women from the pediatric cohort. Atrial and ventricular septal defects were the main birth defects (7/10), but these were not correlated with the use of zidovudine or other antivirals during pregnancy.

The limits of the study are determined by the small sample size and the retrospective design of the study over a long period of time, when the available protocols of antiviral treatment have been modified. The retrospective evaluation of birth defects for HIV-exposed newborns was not based on a single national rule for treatment, surveying and notification. The influence of nonantiretroviral drugs and nutritional supplements on the birth defects were not obtained in this study. No data were available on miscarriages and stillbirths.

## 5. Conclusions

We achieved the aim of preventing new perinatal HIV transmissions, but the rates of mortality and birth defects are over the level of other reported studies. Intensifying HIV screening and pregnancy surveillance are required for earlier HIV diagnosis and the prevention of preterm delivery, neonate death and birth defects. The hope of a new generation free of HIV from parents grown up with HIV is real, but we should provide long time careful pediatric follow up and support. Prevention of HIV perinatal transmission is a continuing challenge, requiring more research to investigate the relationship between pregnancy, antiretroviral history and the long-term progression of HIV in fertile women from pediatric cohorts.

## Figures and Tables

**Figure 1 jcm-11-05020-f001:**
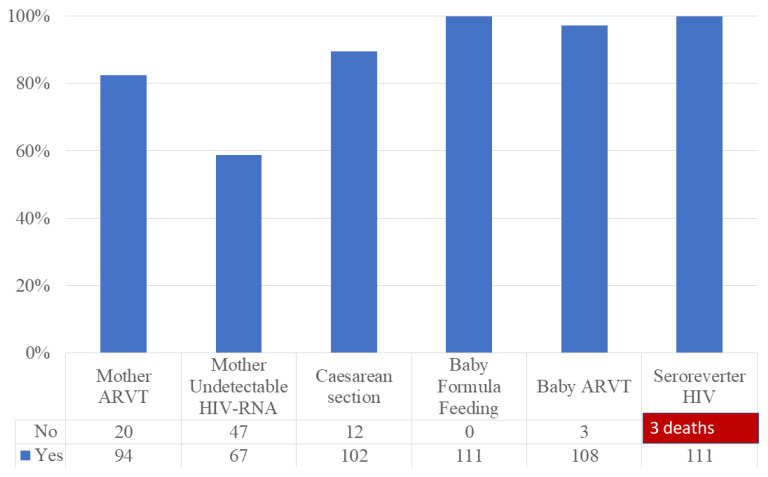
The cascade of care to prevent mother-to-child HIV transmission.

**Table 1 jcm-11-05020-t001:** Data on mothers with HIV infection who gave birth to live children: 2005–2019.

Data	Categories	*N*	%
Living area	Urban	56	49%
	Rural	58	51%
Formal education level	Illiteracy	14	12.28%
	4 years	21	18.42%
	8 years	62	54.38%
	≥12 years	17	14.91%
Smoking	Yes	76	66.66%
	No	38	33.33%
Civil status	Single	16	14%
	Unmarried couple	46	40%
	Married	52	46%
Father HIV status	Positive	39	34%
	Negative	61	54%
	Unknown	14	12%
HIV Diagnostic HIV related to pregnancy	Before pregnancy	92	81%
	During pregnancy	12	10%
	At birth	10	9%
The rank of birth	First birth	73	65%
	Second birth	27	24%
	≥3 births	10	9%
	Twins (2 pairs)	2	2%
Hepatitis B virus co-infection	Yes	32	28%
	No	82	72%
AIDS	Yes	64	57%
	No	49	43%
RNA-HIV(cut-off for detection 400 copies/mL)	Detectable	39	34%
	Undetectable	67	59%
	Unavailable	8	7%
Antiretrovirals during pregnancy	Yes	94	82.45%
	All the pregnancy	70	61.40%
	After the 1st trimester	24	21.05%
	No	20	17.54%

Legend: HIV: Human Immunodeficiency Virus; AIDS: Acquired Immuno-Deficiency Syndrome; RNA: ribonucleic acid.

**Table 2 jcm-11-05020-t002:** Anthropometric characteristics of perinatal exposed newborns.

	Average	SD	Median	Max	Min	CI 0.95	*p* *
Gestational age (weeks)	37.21	2.20	38	41	29	36.8; 37.6	<0.001
Apgar score	8	1.02	8	10	4	7.80; 8.19	<0.001
Weight (grams)	2739	512.03	2800	3800	1300	2635; 2830	<0.001
Length (cm)	47.91	3.32	49	54	33	47.2; 48.5	<0.001
Cranial circumference (cm)	32.60	2.31	33	48	25	32.0; 33.0	<0.001

* Test of the mean (*t*-test); SD: standard deviation; CI: confidence interval.

**Table 3 jcm-11-05020-t003:** Birth defects for newborns of HIV-positive mothers.

Anomaly Group	Findings at Birth	*N*	Study Prevalence %	EUROCAT Prevalence per 10,000 Births All Registries
Cardio-vascular	Atrial septal defect	4	3.50%	14.68
	Ventricular septal defect	3	2.63%	35.83
Urogenital	Hydronephrosis	1	0.87%	11.76
	Hypospadias	1	0.87%	16.18
Abdominal wall	Gastroschisis	1	0.87%	1.14
Overall		10	8.7%	188.23

## Data Availability

Not applicable.

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
