# Peer review of "A Real-Life Action toward the End of HIV Pandemic: Surveillance of Mother-to-Child HIV Transmission in a Center from Southeast Romania"

_jcm, 2022, doi:10.3390/jcm11175020_

Round 1

Reviewer 1 Report

In the reviewed manuscript, Manuela Arbune colleagues report a single centre study at the the Clinical Hospital for Infectious Diseases in Galati, at from South-East of Romania to assess the effectiveness of the Preventing the mother-to-child HIV transmission program.  The ms presents a retrospective study that includes all new-borns of HIV positive mothers from 2005 to 2020 and were followed-up until age of two. The transmission rate from HIV positive mothers to living children has been zero, but neonatal mortality, preterm birth and birth defects still found to be high, mainly for heart damage.

Minor comments/suggestions

-          Introduction segment; the authors should add info re the U=U statement (“the 3rd 95” goal), as in the U=U era, we know that viral suppression due to effective ART constitutes the most effective strategy for preventing onward transmission of infection and, thus, expansion of the HIV epidemic [e.g PARTNER, Opposites Attract, HPTN052 studies etc]

-          Discussion segment; to my personal opinion is quite extensive and not easy to follow; I would suggest to cut some repeated info and make it more concise

-          The ms warrants to be revised by an English native author in order to be significantly improved and would be more easy to read and follow

Author Response

Thank you for the pertinent and useful comments and recommendations.

Reviewer 2 Report

This paper presents very interesting and important data  in the field of PMTCT. Your findings in this cohort of mainly treatment experienced women living with HIV are encouraging (no HIV transmission to the infants).

I found some of the content difficult to follow and I have made comments in the attached file

Author Response

Response to Reviewer 2 Comments

Thank you for the pertinent and useful comments and recommendations.

Point 1:

Introduction segment; the authors should add info re the U=U statement (“the 3rd 95” goal), as in the U=U era, we know that viral suppression due to effective ART constitutes the most effective strategy for preventing onward transmission of infection and, thus, expansion of the HIV epidemic [e.g PARTNER, Opposites Attract, HPTN052 studies etc]

Response 1:

R 54-55: The paragraph: “Antiretrovirals medication have unquestionable benefits in preventing perinatal HIV transmission, but the risk of genotoxicity is not yet sufficiently known.”

Was completed and reformulated

“Antiretrovirals medication have unquestionable benefits in preventing HIV transmission, considering that the goal of treatment is to rich undetectable viraemia, that is equivalent to untransmissibility, meaning “U=U” [Cohen, 2020; NAM 2022], Although the effective treatment constitues the most successful strategy for preventing onward transmission of infection and expansion of HIV epidemic, including mother-to-child transmission, the risk of genotoxicity is incompletely known.”

Supplementary citations:

Cohen MS, Chen YQ, McCauley M, Gamble T, Hosseinipour MC, Kumarasamy N, et al. Antiretroviral therapy for the prevention of HIV-1 transmission. N Engl J Med. 2016;375(9):830–9.

Cohen MS, Gamble T, McCauley M. Prevention of HIV Transmission and the HPTN 052 Study. Annu Rev Med. 2020 Jan 27;71:347-360. doi: 10.1146/annurev-med-110918-034551. Epub 2019 Oct 25. PMID: 31652410.

NAM aidsmap NAM endorses undetectable equals Untransmittable (U=U) consensus statement, 2017. Available: http://www.aidsmap.com/news/feb-2017/nam-endorses-undetectable-equals-untransmittable-uu-consensus-statement#_ednref1 [Accessed 25 JUL 2022].

Point 2:

Discussion segment; to my personal opinion is quite extensive and not easy to follow; I would suggest to cut some repeated info and make it more concise.

Response 2:

We have introduced subtitles for easier follow the sections of discussions.

We have removed the paragraphs about the respiratory distress syndrome in neonates and about minor birth deffects and we have reformulated the paragraph about peculiarities of mother survivor from paediatric cohort.

Local condition of perinatal transmission

The elimination of perinatal transmission secures a AIDS-free born new generation of children. The yearly number of Romanian new-borns in the whole country with perinatal HIV infections has varied in the last 10 years from 6 to 28 cases, meaning between 0.7% and 4.6% from the new cases yearly, distributed by the transmission ways [7]. The perinatal transmission rate reported in other Romanian centres centers varied between 3% and 7% [13,14,15]. The result of our study was “zero maternal-foetal infections”, strengthening the regional hope of eradication HIV infection. Moreover, the natural desire of HIV-positive women to become mothers and to have a family could fulfil their human existence that improve their quality of life.

Peculiarities of HIV positive mother survivor from pediatric cohort

A peculiarity of our study is the preponderance of mothers with paediatric HIV infections associated with the medical care acquired in the first years of life, who survived and gave birth to a new generation of children. Along with the positive emotional aspects, the management of these cases brings to attention two major challenges. Frequently low education and professional qualification, social discrimination including difficulties for job employment, untrust, psychological frustrations or poverty are a huge burden added to the medical history of AIDS and non-AIDS diseases for the young women grown up under all this pressure.

Low education, social discrimination, psychological frustrations, or poverty, as well as AIDS and non-AIDS associated diseases are more frequent burdens for the young females with HIV from pediatric cohorts. Therapeutical fatigue, poor adherence and virologic failure are often difficult to be managed, as other queries with incomplete answers. What is the influence of HIV infection, or maternal antiretroviral medication and caesarean section on preterm birth and birth mortality of mother and child? How efficient is the antiretroviral prophylaxis in new-borns of mothers with long-term therapeutic experiences and possible HIV resistance mutations? What are the implications of antiretroviral therapy on embryo-foetal development and presence of birth defects?

The impact of HIV on maternal and neonatal mortality

HIV-associated global maternal mortality has declined over the past two decades, although in some disadvantaged regions the risk of maternal deaths in HIV-positive women remains more than 5 times higher than in uninfected women [16,17,18]. No maternal deaths were recorded in our centre center. At the same time, WHO statistics for Romania indicate zero maternal deaths associated with HIV and an overall maternal mortality rate with a declining trend, up to 5.2 / 100,000 births [16]. The perinatal mortality rate declined worldwide and the discrepancy between children born to HIV-positive and HIV-negative mothers has been narrowed [19]. During the study period, the general perinatal mortality rate decreased in Romania, from 1.06 to 0.57% [16]. In our study, 3 new-borns died in the first 6 days of life, meaning 2.6% perinatal mortality associated with HIV exposure. This is higher than other recent reports, as the cohort study from UK and Ireland, with 0.41% mortality rate [4].

The impact of preventive interventions on HIV exposed child

Paediatric antiretrovirals were provided in 108 of 111 new-borns, with Zidovudine alone or in combination with Lamivudine ± Nevirapine. The effectiveness of this medications for children of mothers from the paediatric cohort may be limited due to viral resistance mutations acquired by mothers’ therapeutic experience. Little is known on the antiviral administration in children of mothers infected perinatal or during early childhood, with huge therapeutic history. A study of 11 pregnant women who had been infected perinatal with HIV found mutations in resistance to zidovudine and lamivudine. However, all their 21 babies were given prophylaxis with these antivirals. Although maternal viremia at birth was detectable in half of the cases, all children were uninfected [20].

Preterm delivery (<37 weeks) could be an additional factor for HIV transmission, although the risk is minimal for women under antiretroviral treatment, supplemented with caesarean delivery, if with predelivery viral load is over less than 1000 copies/mL [5, 8, 21]. The rate of preterm births to HIV exposed babies in our study was 28.07%, that is 2.7 times higher than an average of 10.6% from global European data [22]. Increased rates of preterm births related to mother HIV infection have been also reported by other studies: 22.4% in another Romanian centre center 22.6% in Italy, 20% in the USA, 21.7% in Brazil [23,24,25,26].

Although elective caesarean section is recommended at 38 weeks of pregnancy, the decisions to achieve it earlier was related to premature beginning of labour labor, premature rupture of the membranes, or evidence of the foetal distress. Considering the high rate of detectable or unavailable mothers’ viral load at birth, the caesarean section is chosen in Romania as a safer procedure than vaginal delivery for the risk of mother to child HIV transmission, than vaginal delivery [14].

Zero maternofoetal infections in living babies were achieved in our study, even if the cascade of care was not fully accomplished. Systematically feeding with baby formula had a major contribution, considering that 5–20% of mother-to-child transmitted infections are through breastfeeding [3]. The weakest indicator is the detectable HIV viral load in 41% of pregnant women, due to either late HIV diagnostic during pregnancy, or non-adherence when HIV diagnostic was known. The reasons for non-adherence could be the fear of disclosure HIV diagnostic, the fear that antiretrovirals harm the baby or therapeutical fatigue. The resistance tests are not usually available in our site, but the acquired resistance to antiretroviral drugs is considered in virologic failure. Caesarean section as usual indication in 92% of births could have decreased the risk of transmission, related to the detectable viral load. Antiretrovirals were provided to all the alive babies in the first hours of life, but there were discontinued in 3 cases, due to toxicity.

The respiratory distress has been diagnosed in nearly a quarter of new-borns of HIV positive mothers. The rate of respiratory distress syndrome in Romania was reported depending on gestational age and type of delivery, from 5% to 65% after caesarean and from 3% to 60% after vaginal birth [27]. Given the national recommendations for elective caesarean delivery in HIV-infected pregnancies, the frequency of respiratory distress is consistent with data from the general population.

The impact of HIV and antiretroviral drugs perinatal exposure on birth defects

Congenital anomalies can be caused by genetic, environmental, or other risk factors, but almost half of the cases cannot be related to a specific cause. Most trials of in utero exposure to antiretrovirals have not shown significant risk for birth defects, but there are studies that show higher risks for some specific drugs. The Antiretroviral Pregnancy Registry (APR) reported 2.87% birth defects, compared to the metropolitan Atlanta congenital programme of 2.72% and 3.56 % live births observed in a cohort of 2527 new-borns of HIV-infected mothers [28,29,30].

Paediatric AIDS Clinical Trials Group 219 and 219 C protocols enrolled 2202 HIV-exposed children before one year of age, reporting higher prevalence of birth defects of 5.3%, with a higher rate of heart defects when zidovudine was administered in the first trimester of pregnancy [31]. The French Perinatal Cohort included 13,124 live births from 1994 to 2010, of which 42% were exposed to ART in the first trimester of pregnancy. A prevalence of 4.4% of birth defects was reported on children prospective followed up until age of two, according to the EUROCAT classification, but with no evidence of association with antivirals exposure as lopinavir, ritonavir, nevirapine, tenofovir, stavudine, or abacavir. Associations between zidovudine exposure in the first trimester with congenital heart defects or to efavirenz with neurological defects, were not significant [32].

The birth defects rate in our study, according to the EUROCAT classification, was 8.7%, higher of 1.52 times in pregnant women from the paediatric cohort. Atrial and ventricular septal defects were the main birth defects (7/10), but there were not correlated with the use of zidovudine or other antivirals during pregnancy. Among the minor defects, which are not included in the EUROCAT classification, the most numerous were haemangiomas, reported in more than a quarter of the new-borns in the study, almost 3 times more frequent compared to the general population. The incidence of infant haemangiomas varies between 3 and 10%, is higher in premature babies and tends to increase in recent decades [33,34,35,36].

.

Point 3:

The ms warrants to be revised by an English native author in order to be significantly improved and would be more easy to read and follow

Response 3:

The manuscript should be final revised for the native language by the journal specialists, according to our application.